# Proteomic and ubiquitinome analysis reveal that microgravity affects glucose metabolism of mouse hearts by remodeling non-degradative ubiquitination

Xin Zhang[1], Xuemei Zhou[1], Zhiwei Tu[1], Lihua Qiang[1], Zhe Lu[2], Yuping Xie[1], Cui Hua Liu[2], Lingqiang Zhang[1]*, Yesheng Fu[1]*

1 State Key Laboratory of Proteomics, National Center for Protein Sciences (Beijing), Beijing Institute of Lifeomics, Beijing, China, 2 Institute of Microbiology (Chinese Academy of Sciences), CAS Key Laboratory of Pathogenic Microbiology and Immunology, Savaid Medical School, University of Chinese Academy of Sciences, Beijing, China

* zhanglq@nic.bmi.ac.cn (LZ); fyshwork2013@126.com(YF)

**Data Availability Statement:** Raw mass spectrometry data generated in this study have been deposited in ProteomeXchange Consortium

## Abstract

Long-term exposure to a microgravity environment leads to structural and functional changes in hearts of astronauts. Although several studies have reported mechanisms of cardiac damage under microgravity conditions, comprehensive research on changes at the protein level in these hearts is still lacking. In this study, proteomic analysis of microgravity-exposed hearts identified 156 differentially expressed proteins, and ubiquitinomic analysis of these hearts identified 169 proteins with differential ubiquitination modifications. Integrated ubiquitinomic and proteomic analysis revealed that differential proteomic changes caused by transcription affect the immune response in microgravity-exposed hearts. Additionally, changes in ubiquitination modifications under microgravity conditions excessively activated certain kinases, such as hexokinase and phosphofructokinase, leading to cardiac metabolic disorders. These findings provide new insights into the mechanisms of cardiac damage under microgravity conditions.

## Introduction

In space exploration, microgravity have multiple impacts on the human body, causing effects including muscle atrophy [1], osteoporosis [2], decreased immune function [3], changes of cardiovascular system [4–7], and other physiological responses [8,9]. For cardiovascular system, prolonged exposure to a microgravity environment during spaceflight leads to structural and functional changes in the heart [10]. Due to the reduced body load in weightlessness, the workload on the heart is correspondingly decreased, which causes myocardial atrophy [11–16]. Studies have shown that long-term exposure to microgravity results in an increase in the left ventricular volume of the heart, a decrease in wall thickness, leading to the relaxation and enlargement of the heart muscle, thereby reducing its

via the iProX repository with the identifier PXD051620 and these data have been publicly released. All other relevant raw data are within Supporting Information files.

**Funding:** This work was supported by the National Natural Science Foundation of China (Project No. 82192881 for Lingqiang Zhang; Project No. 32201023 for Yesheng Fu), the National Key Research and Development Program of China (Project No. 2021YFA1300200 for Lingqiang Zhang), and the Young Elite Scientists Sponsorship Program by the China Association for Science and Technology (Project No. YESS20220049 for Yesheng Fu). The funders had no role in study design, data collection and analysis, decision to publish, or preparation of the manuscript.

**Competing interests:** The authors have declared that no competing interests exist.

contractile strength and pumping ability [17,18]. These changes result in symptoms such as dizziness and palpitations in astronauts upon return to Earth, and even increase the risk of heart disease [19,20]. However, the adaptive responses and functional changes of the heart under microgravity conditions are complex, and mechanism studies are needed to further elucidate the effects of microgravity on the heart.

There have been several studies on the mechanisms of heart damage exposed to microgravity [21,22]. RNA sequencing (RNA-seq) analysis of the hearts of fruit flies sent into space reveals a reduced expression of myocardial/extracellular matrix (ECM) genes and a significant increase in proteasome gene expression [23]. Canonical gene deletion of *Capns1* shows that the effect of unloading on cardiomyocytes is alleviated by attenuating ERK1/2 and p38 signaling [24]. Recently, it has been reported that the 3'-UTR of Ckip-1 attenuates simulated microgravity-induced cardiac atrophy [25]. The calcium/calmodulin-dependent protein kinase II (CaMKII)/histone deacetylase 4 (HDAC4)/myocyte-specific enhancer factor 2C (MEF2C) axis, which is a critical regulator of pressure overload-induced cardiac remodeling, is also found to be regulated by WW domain-containing E3 ubiquitin protein ligase 1 (WWP1) [26]. Although these mechanistic studies focus on the pathophysiology of the heart under weightlessness, systematic omics studies, especially on the overall protein level in cardiac tissues under weightlessness, are still lacking.

Proteomic omics technologies, including proteomics and ubiquitinome, are the useful tools for analyzing protein expressions and regulatory pathways induced by microgravity [27,28]. Proteomics, through high-throughput techniques such as mass spectrometry, reveals the expression levels, modification states, interactions, and localization of proteins in organisms, providing a comprehensive understanding of biological processes within organisms [29–33]. Ubiquitination modification is a sensitive indicator of cellular responses to environmental changes and stress, and thus serves as an important indicator for assessing the impact of microgravity on cell function and adaptability [34–36]. Here, we integrated proteomic and ubiquitinomic analyses of hearts under microgravity conditions and found that proteomic changes were closely associated with biological functions such as immune response, RNA splicing, and protein folding. In contrast to the proteome, ubiquitinomic changes tended to be more related to biological functions associated with muscle contraction and glucose metabolism. These findings would strength the understanding of microgravity-induced cardiac damage and the underlying molecular mechanisms.

## Materials and methods

### Animal model

Male C57BL/6 mice were purchased from Vital River (Beijing) and maintained in accordance with protocols approved by the Beijing Institute of Lifeomics. All mice were kept in a 12-hour light/12-hour dark cycle. The mice had free access to standard chow and water. The animal room was maintained at 24˚C and 50% humidity. Trained members of the research team monitored the food and water consumption and overall health status of all mice daily, and no adverse conditions or health issues were noted. No animals died before meeting the criteria for euthanasia. After the mice were acclimatized for two weeks, the experiments commenced. During the course of the experiments, the mice were 12 weeks old and age- and sex-matched in each experiment. All experimental procedures in mice were approved by the institutional animal care and use committee at the Beijing Institute of Lifeomics (approval number: IACUC-20221231-79MBe). After the recordings were completed, all mice were sacrificed by cervical dislocation.

## Simulated microgravity for mice

Male C57BL/6J mice, aged 12 weeks with an average weight of 26 ± 1.5 grams, were individually housed in custom cages equipped with a tail suspension system to create hindlimb unloading (HU) conditions. The suspension angle was adjusted to approximately 30° from the horizontal to prevent the hind limbs of mice from contacting any supporting surface, ensuring that their forelimbs could be used for movement. Mice in the HU group were subjected to this condition for 40 days, while the Sham group was housed under identical conditions without tail suspension. All mice were maintained under standard animal housing conditions with a 12-hour light-dark cycle and had ad libitum access to food and water.

## Total protein extraction and FASP enzymatic digestion

The heart tissue of mice was extracted and grinded in liquid nitrogen. Freshly prepared SDC lysis buffer (1% SDC (sodium deoxycholate, Ron Reagent, R096950), 10mM TCEP (Tris(2-carboxyethyl) phosphine hydrochloride, Sigma, C4706), 40mM CAA (2-chloroacetamide, Sigma, 22790)) was added and the lysates were incubated at 4°C for 2 hours. Next, the lysates were heated at 95°C for 10 minutes and ultrasonicated with an ultrasonic disruptor (Xinzhi Biology, SCIENTZ-IID). The resulting samples were added to an ultrafiltration unit (Millipore, UFC5030BK-30KD), and washed for three times with 50mM ABC buffer (ammonium bicarbonate, Sigma, A6141). Then, the samples are incubated in trypsin solution at 37°C for 16 hours (overnight), following with TFA (Trifluoroacetic acid, ACROS, 139721000) to a final concentration of 0.3% for neutralization, and freeze-dry at 4°C.

## Enrichment of K-ε-GG

Dissolve the peptide segments in 1.4ml IAP (Inhibitor of Apoptosis Proteins, 50 mM MOPS (3-(N-Morpholino) propanesulfonic acid, Sigma, M5162, pH 7.2), 10 mM Na2HPO4, 50 mM NaCl), sonicate, and then add pre-treated K-ε-GG beads. Seal and rotate at 4°C for 2 hours. Add 55 μl of 0.15% TFA for elution, collect the supernatant, elute twice, and combine the eluates. After desalting with a Tip desalting column (Thermo Scientific, 87784), elute the peptide segments with 0.1% TFA. After freeze-drying, the sample can be identified by mass spectrometry.

## Mass spectrometry identification

Freeze-dried sample was dissolved in 0.1% formic acid (Sigma, 5.33002) and an equivalent of 2 μg peptides was loaded onto a self-packed C18 analytical column (1.9um particle size, 100 Å pole size, 30 cm x 150 μm) using an EASY-nLC 1200 UPLC system. A 120 min LC gradient was applied for samples of proteome analysis. The eluted peptides were sprayed into a spray ion source, attached to a Q Exactive HF mass spectrometer (Thermo Fisher Scientific) scanned in DIA (Data Independent Acquisition) mode. MS1 was scanned at a m/z range of 400 ~ 1200 with a resolution of 120,000 at m/z 200 and AGC (Automatic Gain Control) target was set to 3E6, with a maximum injection time of 80 ms. Thirty-two ms2 scans with a fixed window width of 26 m/z were followed with HCD energy of NEC 27%. The ms2 scan was performed at a resolution of 30,000 at m/z 200, with an AGC target of 3E6, a maximum injection time of 41 ms, and a default charge state of 4. Instead, A 90 min LC gradient was applied for samples of lysine ubiquitination and ms1 scanned at a m/z range of 400–1650. Thirty-three ms2 scans with a fixed window width of 25 m/z were followed.

## Database search and bioinformatic analysis

The database search of mass spectrometry raw data against Uniport mouse proteome fasta database was preformed using Spectronaut (Biognosys, Switzerland, version 16.1). The

digestion enzyme was Typspin/P with a maximum of two missed cleavages. Cysteine carbami-domethylation was set as a fixed modification, methionine oxidation and acetylation of the protein N-terminus were set as variable modifications with a maximum of five modification sites per peptide. False discovery rate (FDR) was controlled to 1% for peptides and proteins. A similar Spectronaut search was performed on the raw data for lysine ubiquitination. Additionally, ubiquitination on lysine was set as an extra variable modification. PTM localization filter and PTM localization cutoff were both set to 0.75.

The Spectronaut precursor-level quantification data was collapsed into a site-level quantification using Perseus with plugin peptide collapse (https://github.com/AlexHgO/Perseus_Plugin_Peptide_Collapse). R software version 4.1.2 (https://www.r-project.org/) was used to process outputs of Spectronaut and Perseus with R/Biocmanger packages tidyverse, DMwR2, pheatmap and limma. Differentially expressed proteins or ubiquitinated sites were defined as having a fold change greater than 1.5 (HU vs Sham) and an adjusted p value less than 0.05, unless otherwise stated. DAVID Bioinformatics Resources tool (version 2021) was used to retrieve Gene Ontology (GO) and Kyoto Encyclopedia of Genes and Genomes (KEGG) pathway enrichment of proteins.

## RNA extraction and quantitative real-time PCR

Total RNA was extracted from cells in mouse hearts with TRIzol reagent (Invitrogen, 10296010) following the manufacturer's instructions. Total RNA was reverse transcribed into cDNA using the ReverTra Ace (ⓡ) qPCR RT Master Mix (Toyobo, FSQ-201), and quantitative real-time PCR (qRT-PCR) was carried out using SYBR (ⓡ) Green Realtime PCR Master Mix (Toyobo, QPK201). Relative fold difference was calculated using the ΔΔCt method. The primers used were as follows:

Ighg2b-F: TTCACAGACGAGGTCTGCAC
Ighg2b-R: TCAAGAGCCATGTGCCTCAG
Ighg2c-F: CCATCGGTCTATCCACTGGC
Ighg2c-R: GTCCACTTTGGTGCTGCTTG
Igkc-F: AGCTAGTGAGATCAGGGGCA
Igkc-R: CGCTGTTTCATCCTCTGGGT
Fga-F: GCCGACCAATGGGAAGAGAA
Fga-R: AGCCCCCACTTTCTAACCCT
Pfkp-F: GAGCGTCCTCTAGCATCCAC
Pfkp-R: TCCAACTAGCAGCAGCTCAC
Aldoa-F: AGGCTGCTCCATCAACACTC
Aldoa-R: GGAAAGAGCCTGAAGACCCC
Pgk1-F: TTGTGCATTGTAGAGGGCGT
Pgk1-R: GACGAAGCTAACCAGAGGCT

## Measurement of PFK and HK activities and PA content

For measurement of PFK and HK Activity, the mouse hearts were homogenized in an ice bath following the addition of the extract. The supernatant was extracted by centrifugation at 10,000 $g$ for 10 min. The Phosphofructokinase (PFK) Activity Assay Kit (Solarbio, BC0535) and Hexokinase (HK) Activity Assay Kit (Solarbio, BC0745) were used to detect PFK and HK activity respectively. The absorbance of samples at 340 nm was measured with the Multiskan FC microplate reader (Thermo Fisher). For measurement of PA content, the mouse hearts were homogenized in an ice bath following the addition of the extract. The supernatant was extracted by centrifugation at 10,000 $g$ for 10 min. The Pyruvate (PA) Content Assay Kit

(Solarbio, BC2205) was used to detect PA content. The absorbance of samples at 520 nm was measured with the Multiskan FC microplate reader (Thermo Fisher).

### Conventional echocardiography

The high-resolution micro-ultrasound system Vevo 2100, manufactured by Fujifilm VisualSonics, equipped with an MS400 (30MHz center frequency) transducer, was utilized for transthoracic echocardiography. Briefly, mice were anesthetized with isoflurane (RWD, R510-22), and hair was shaved from the neck to the mid-abdominal region. Subsequently, mice were placed on a heated table to maintain a core body temperature of 37˚C, in a supine position with built-in electrocardiogram leads. Anesthesia was maintained with 1%-3% isoflurane throughout the imaging procedure. B-mode and M-mode images were acquired from the short-axis view of the aorta to assess heart rate, fractional shortening (FS), and ejection fraction (EF).

### The protein purification and ubiquitinated protein capture experiment

6×His-Halo-TUBE were expressed in BL21 (DE3) and purified through the HisTrap HP column (Cytiva). The supernatants of each sample were incubated with the pre-loaded TUBE-HaloLink resin overnight. The resins were washed three times in HEPES buffer by centrifuging at 1,000 g for 10 minutes. The remaining proteins were released from the resin via boiling in loading buffer. The TUBE-captured samples were loaded onto SDS-PAGE, and immunoblot with indicated antibodies.

### Statistical analysis

Data are represented as mean ± SEM from at least three independent experiments. The quantified data with statistical analysis were performed using GraphPad Prism 9 software (v9.5.1). The significance of difference was evaluated using the two-tailed unpaired Student's t test. $p$ value < 0.05 was used to indicate significance. $p$ values are indicated as follows: $^*p < 0.05$, $^{**}p < 0.01$, $^{****}p < 0.0001$ and n.s., no significance.

## Results

### Proteomic analysis of cardiac responses to simulated microgravity

In order to study the protein-level changes of hearts stimulated by microgravity, a tail suspension model that mimics microgravity conditions is implemented [37]. The cardiac functions of Sham and hindlimb unloading (HU) mice were monitored by heart rates, fractional shortenings (FS), and ejection fractions (EF). The results showed a significant decrease in both FS and EF in HU mice with heart rate between 450 and 520 beats per minute (S1 Fig), indicating a notable decline in cardiac functions of HU mice compared to the control group. To further unravel the proteomic changes in these microgravity-stimulated hearts, quantitative mass spectrometric analysis of both the proteome and ubiquitinome were conducted (Fig 1A). The repeatability analysis of proteomic data showed that there was a good consistency between repeated experiments. (Fig 1B). Further statistical analysis revealed that microgravity stimulated 90 upregulated proteins and 66 downregulated proteins out of the screened 4111 proteins (Fig 1C and 1D and S1 Table). To better understand the impact of microgravity on cardiac biological functions, Gene Ontology (GO) enrichment of these proteins was analyzed. We found that upregulated proteins were closely associated with biological functions such as immune response, RNA splicing, and protein folding (Fig 1E). In contrast, downregulated proteins were associated with cellular cytoskeleton, lipid metabolism, and energy metabolism. This suggested that several essential biological processes were affected by microgravity.

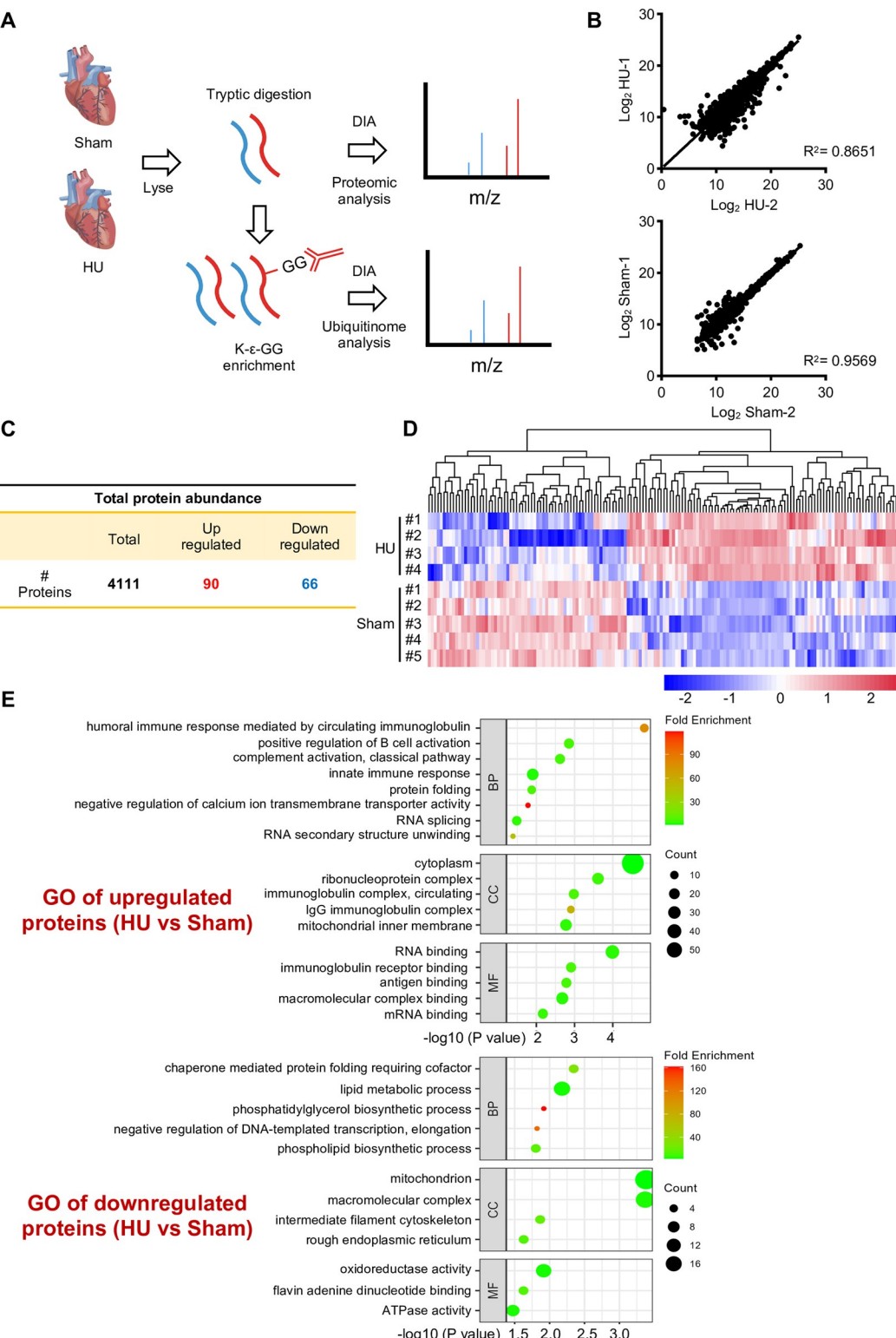

**Fig 1. Proteomic analysis of cardiac responses to simulated microgravity.** (A) Sham and hindlimb unloading (HU) mice were used in this study. Cardiac tissues were collected from these mice and processed into tissue lysates for proteomic and ubiquitinome mass spectrometry identification respectively. (B) The proteome correlation of the Log2 HU (or Log2 Sham) ratio was analyzed between replicates, n = 4111 per group. (C) The chart presenting the total number of identified proteins. Among them, 90 proteins were significantly upregulated, and 66 proteins were significantly downregulated. The differential analysis was performed with FC (foldchanges) > 1.5 and p values < 0.05. (D) A heatmap was generated to

illustrate the differences in proteome levels between the hindlimb unloading (HU) and sham mice groups. (E) Functional enrichment analysis was performed on differential proteins (FC > 1.5 and p value < 0.05), including upregulated and downregulated proteins.

## Proteomic analysis of cardiac ubiquitinome under microgravity conditions

Next, we evaluated the ubiquitinome data and also found similarly good consistency between Sham and HU groups (Fig 2A). Statistical analysis of ubiquitination sites in these two groups revealed an average of 1442 ubiquitination sites per mouse (Fig 2B). Ubiquitination was not uniformly distributed among peptides of different lengths [38]. Our data revealed that approximately 1 out of 4 ubiquitination sites in mouse cardiac tissue proteins were single-site modifications, with fewer proteins having modifications at more than 100 sites (Fig 2C). Further analysis revealed 99 proteins with upregulated ubiquitination modifications and 70 proteins with downregulated ubiquitination modifications in HU groups (Fig 2D and S2 Table). Subsequent heatmap analysis of these proteins revealed that, unlike proteomic data, ubiquitinome data tended to be associated with changes in muscle contraction and glucose metabolism (Fig 2E and S2 Table). GO analysis also yielded similar conclusions, with upregulated or downregulated ubiquitinated proteins being closely associated with muscle contraction, endocytosis, and glycolysis (Fig 2F). We further analyzed proteins involved in endocytosis regulation, such as Ehd4, Ehd2, and Picalm, and found a significant upregulation trend in ubiquitination modifications with microgravity stimulation (Fig 2G). Conversely, proteins associated with ATPase activity, such as Atp1b1, Tnnt2, and Tpm1, showed a significant downregulation trend in ubiquitination sites under microgravity condition (Fig 2H). Overall, these results revealed widespread and varied ubiquitination modifications in microgravity exposed hearts.

## Differential proteomic changes regulated by transcription affect immune response in microgravity exposed hearts

Significant changes in proteomics and ubiquitinome analysis of microgravity-exposed hearts prompted us to explore the potential relationship between them. Comparative analysis identified 448 proteins detected in both proteomic and ubiquitinome data (Fig 3A and S3 Table). Proteins with changes at the protein level but not regulated by ubiquitination in proteomic data were considered to be potentially regulated by mRNA. Analysis of ubiquitinome data showed that 14 proteins were potentially mRNA-regulated (protein levels changed, ubiquitination levels unchanged) (Fig 3B and S4 Table). Meanwhile, analysis of proteomic data found that 454 differential proteins were ubiquitination-related proteins, and 154 proteins were potentially regulated by mRNA (proteins upregulated or downregulated, ubiquitination unchanged) (Fig 3C and S5 Table). Further GO analysis of these potentially mRNA-regulated proteins revealed that their enrichment was mainly in immune response regulation (Fig 3D). Among 90 proteins potentially upregulated by mRNA, we randomly selected four significantly upregulated proteins for experimental validation. The results indicated that the mRNA levels of these proteins were significantly upregulated (Fig 3E and 3F). Thus, we found a certain correlation between proteomic and ubiquitinome changes under microgravity conditions, with several proteomic changes regulated at the transcription level and closely related to immune response regulation.

## Differential analysis of ubiquitinome reveals potential effects on glucose metabolism under microgravity conditions

Subsequently, we analyzed the levels of ubiquitin chain types in ubiquitinome data and found no significant overall changes in the ubiquitin chain types (Fig 4A), suggesting that

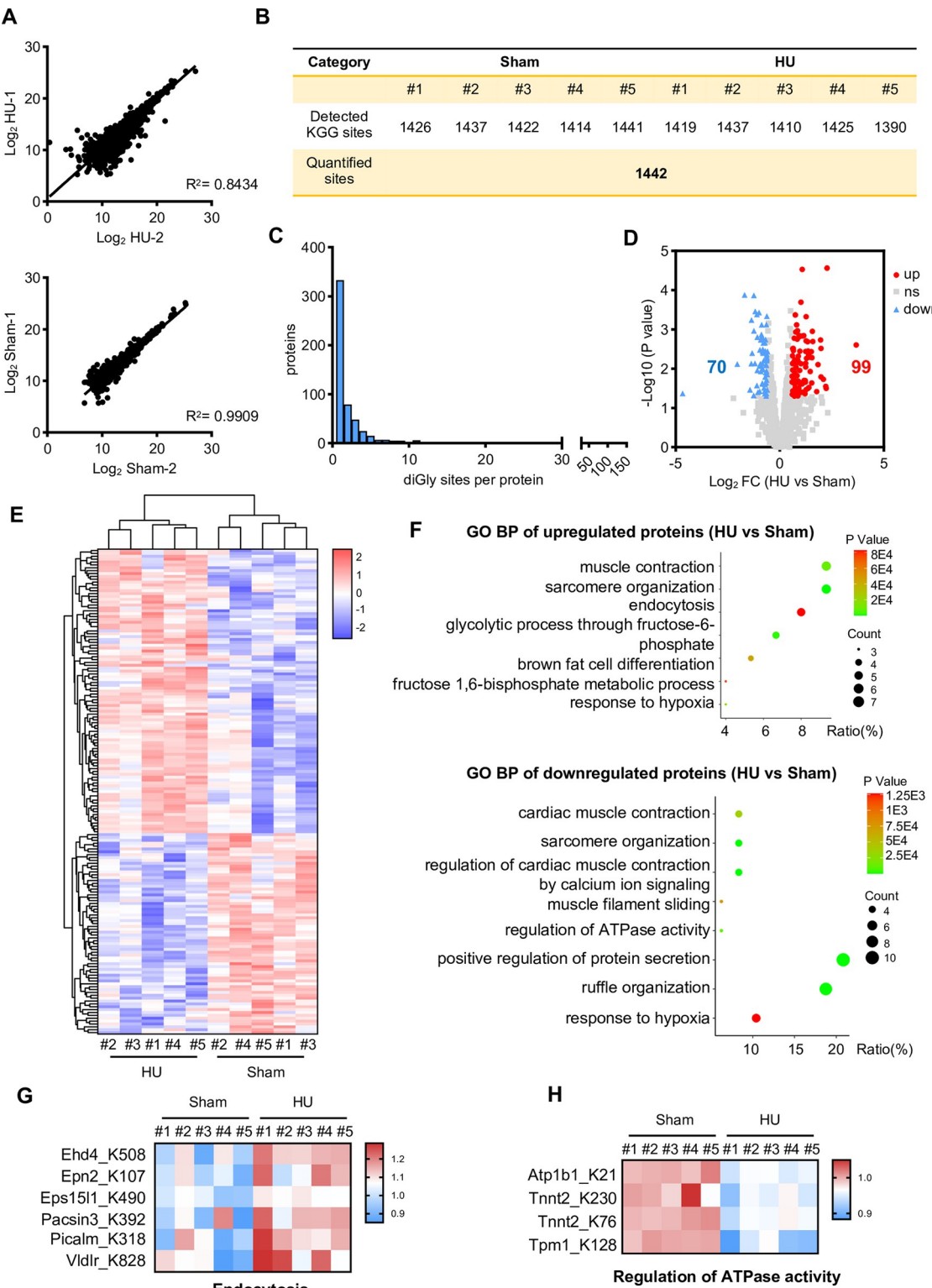

**Fig 2. Proteomic analysis of cardiac ubiquitinome under microgravity conditions.** (A) The correlation of the ubiquitinome in terms of the Log2 HU (or Log2 Sham) ratio between replicates, n = 1442 per group. (B) The chart presenting the total number of identified ubiquitination sites. (C) The distribution of ubiquitinated proteins based on the number of ubiquitination sites. (D) A volcano plot depicting changes in protein ubiquitination levels in mice following tail suspension. Red dots represent 99 upregulated ubiquitination sites out of the screened 1442 sites, while blue dots represent 70 downregulated ubiquitination sites, with a p-value of

0.05 and Foldchange > 1.5. (E) A heatmap illustrating differences in ubiquitination levels between the hindlimb unloading (HU) group and the sham mice group, with five mice in each group. (F) Gene Ontology (GO) analysis determining the biological functions affected by upregulated and downregulated ubiquitinated proteins (HU vs Sham). (G) A heatmap analysis highlighting differences in ubiquitination levels of proteins related to endocytosis between hindlimb unloading (HU) and sham mouse cardiac tissues, n = 5 per group. (H) A heatmap analysis showing differences in ubiquitination levels of proteins regulating ATP kinase activity between hindlimb unloading (HU) and sham mouse cardiac tissues, n = 5 per group.

weightlessness did not specifically increase or decrease a certain ubiquitin chain type. Proteins with changes at the ubiquitination level are always considered as substrates of ubiquitination. Among 545 potential ubiquitinated protein, most of them showed no change in both protein and ubiquitination levels, and only 108 proteins were non-degradable substrates (ubiquitination levels changed, protein levels unchanged) (Fig 3B and S4 Table). Subsequent GO analysis of these 108 non-degradation type ubiquitination-modified proteins showed that these proteins were closely related to muscle contraction, oxidative stress, and glucose metabolism (Fig 4B). We next selected glucose metabolism pathway for further analysis and found that the ubiquitination levels of most regulatory proteins exhibited significant upregulation or downregulation (Fig 4C). Those proteins showing significant changes, such as PFKP, ALDOA, and PGK1, are important enzymes in glucose metabolism [39], indicating that ubiquitination modification under weightless conditions probably participated in regulating glucose metabolism. To verify this, we selected PFKP, which showed the most significant increase in ubiquitination levels among the screened kinases, for validation. We found that under weightlessness conditions, more ubiquitin chains were enriched on PFKP proteins (Fig 4D). Meanwhile, mRNA levels of PFKP, ALDOA, and PGK1 showed no significant changes (Fig 4E). Non-degradation type ubiquitination is always involved in protein trafficking, kinase and phosphatase activation [40,41]. Indeed, the cardiac biochemical indicators related to glucose metabolism were detected, and the results showed that the activities of hexokinase (HK) and phosphofructokinase (PFK) as well as the content of pyruvic acid (PA) were significantly elevated in microgravity-exposed hearts (Fig 4F). Studies have shown that the excessive activation of important kinases involved in glycolysis causes excessive activation of glucose metabolism, which interferes with normal metabolic signaling pathways, affects the energy perception and metabolic regulation of cardiomyocytes, and finally damages heart functions [42–44]. Therefore, our data indicated that the increase in ubiquitination modifications under weightless conditions promoted excessive kinase activations, leading to metabolic disorders in hearts, and providing a potential explanation for microgravity-induced cardiac dysfunction.

## Discussion

Using the tail suspension model in mice to simulate weightlessness, we conducted quantitative mass spectrometry analysis and obtained proteomic and ubiquitinome data of mouse hearts under weightless conditions. Proteomic analysis revealed that overall protein changes in microgravity-exposed hearts tended to cause alterations in cellular cytoskeletal structure, lipid metabolism, immune response, and energy metabolism. Meanwhile, ubiquitinomic analysis showed that non-degradative ubiquitination modifications mainly regulated various kinases in the glucose pathway under weightlessness.

Literature reports have indicated that the NF-κB pathway-associated immune process is significantly activated in mice under microgravity conditions [45]. Consistent with this, our cardiac proteome analysis under microgravity conditions identified a set of proteins significantly enriched in immune pathways. Subsequent analysis of these proteins revealed that the transcription levels of several immunoglobulins were markedly elevated, indicating that the molecular mechanism underlying immune system diseases induced by microgravity may be related

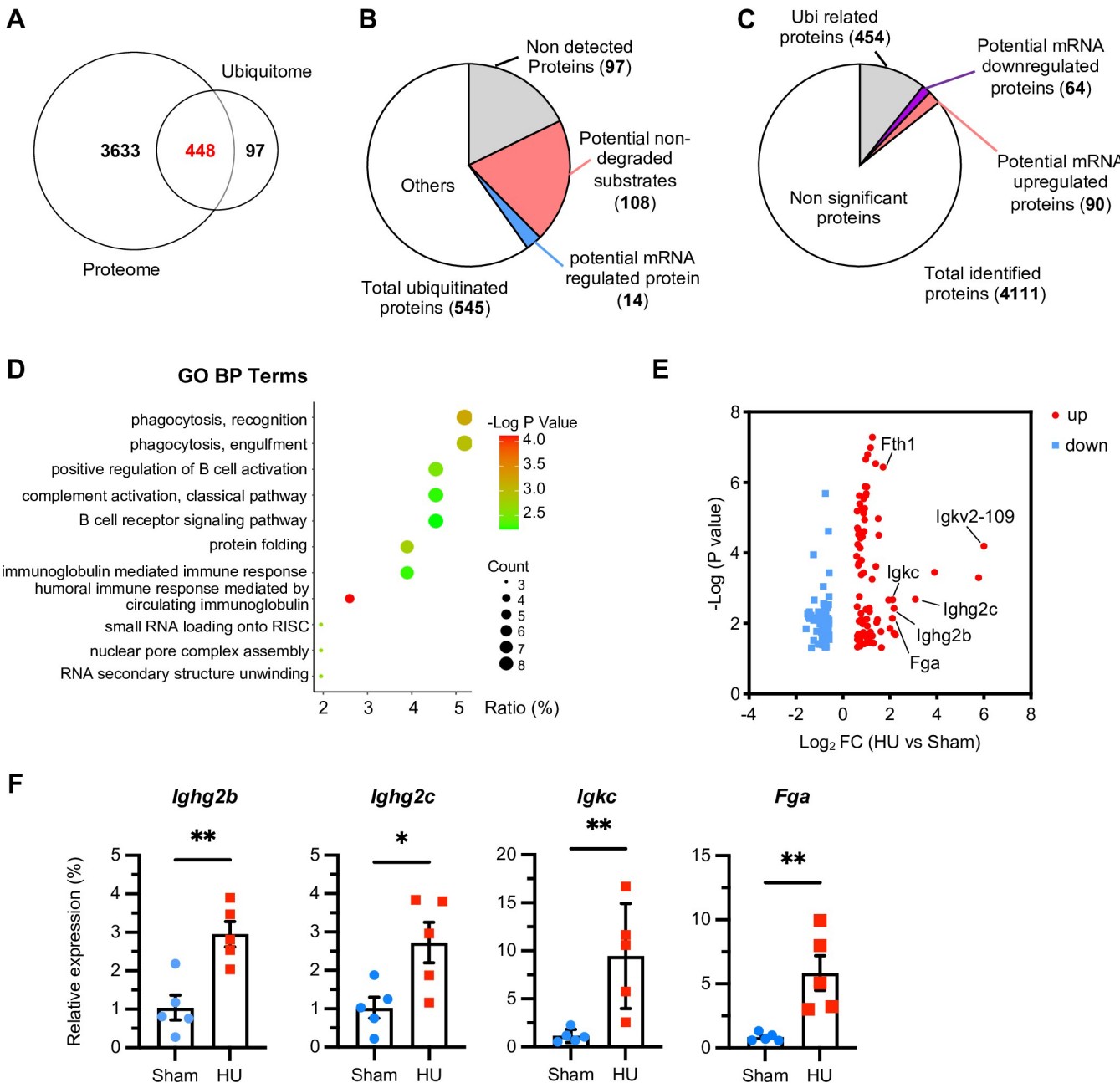

**Fig 3. Differential proteomic changes regulated by transcription levels affect immune response in organism.** (A) A Venn diagram illustrating the overlap in the number of proteins identified through proteomic and ubiquitinome analyses. (B) A detailed pie chart showing the distribution of proteins regulated by different levels in the proteomic data. (C) A detailed pie chart showing the distribution of proteins regulated by different levels in the ubiquitinome data. (D) Biological process terms derived from Gene Ontology (GO) analysis of potentially mRNA-regulated proteins were highlighted. (E) A volcano plot displaying the proteins that were potentially regulated on transcriptional levels. Red dots represent upregulated proteins, while blue dots represent downregulated proteins, with a p-value of 0.05 and Foldchange > 1.5. (F) Quantitative PCR (qPCR) validating the proteins that were potentially under transcriptional regulation. n = 5 mice per group. Data were presented as mean ± SEM. $^*p < 0.05$, $^{**}p < 0.01$.

to immunoglobulins-related processes. However, the regulatory role of immunoglobulins such as Ighg2b and Ighg2c on cardiac inflammatory responses under microgravity requires further study in the future.

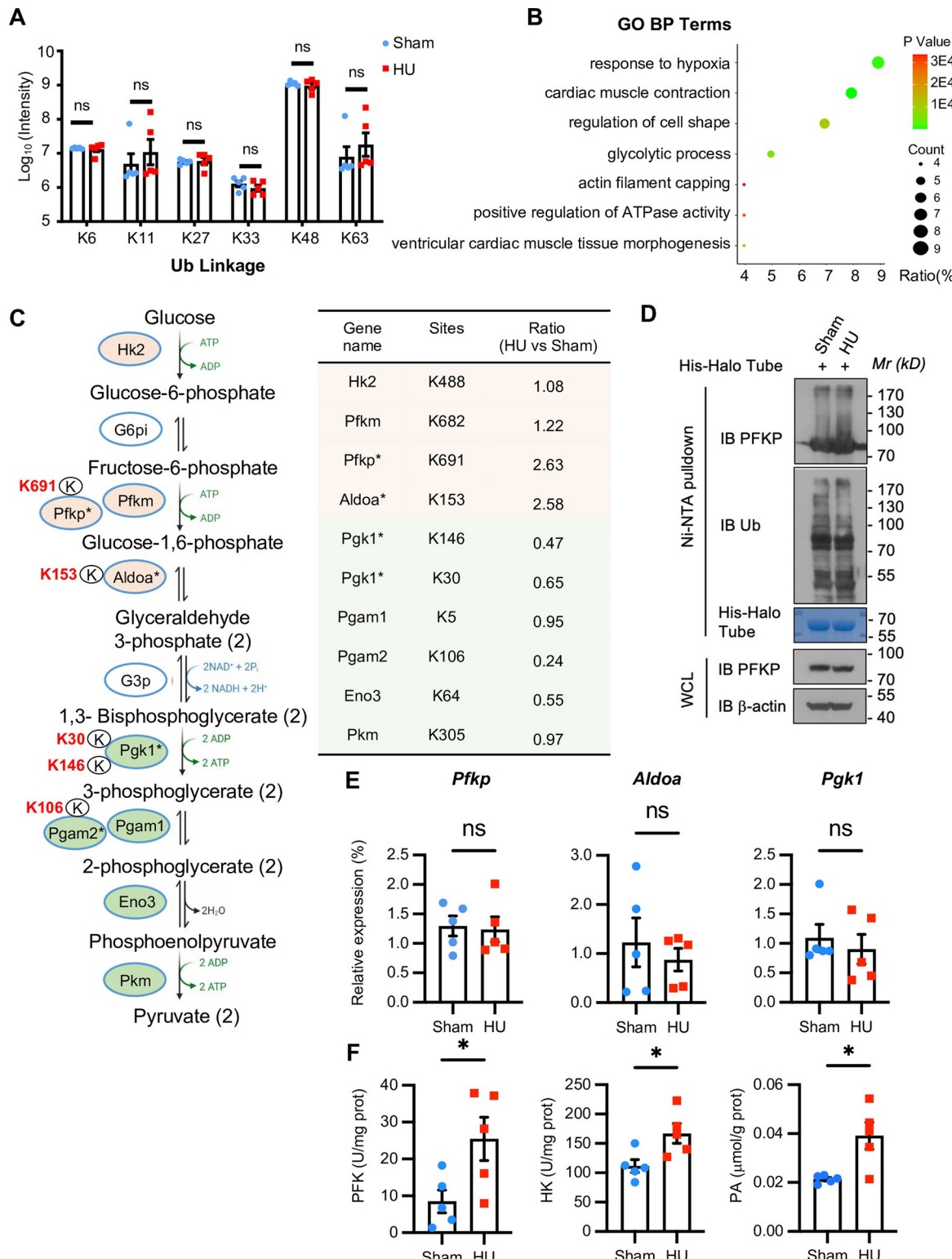

**Fig 4. Differential analysis of ubiquitinome reveals potential effects on glucose metabolism under microgravity conditions.** (A) A comprehensive analysis showing the differences in ubiquitin chain types of cardiac tissues from hindlimb unloading (HU) and sham mice, n = 5 per group. (B) Gene Ontology (GO) analysis showing biological functions that were potentially affected by the non-degradative ubiquitination-modified proteins. (C) Pathway diagrams illustrating the significantly differential changes in ubiquitination levels of multiple glycolysis-related proteins in cardiac tissues of hindlimb unloading (HU) mice. The red areas represent proteins with

relatively upregulated ubiquitination levels, and the green areas represent proteins with relatively downregulated ubiquitination levels. The asterisks represent p value < 0.05 (HU vs Sham), n = 5 per group. (D) The ubiquitination levels of PFKP were enriched by the his-halo-TUBE pulldown in hindlimb unloading (HU) and sham mice. (E) Quantitative PCR (qPCR) detecting changes in the transcriptional levels of glycolysis-related proteins in hindlimb unloading (HU) and sham mice. n = 5 mice per group. Data were presented as mean ± SEM and ns, no significant. (F) Biochemical changes of phosphofructokinase (PFK), hexokinase (HK), and pyruvic acid (PA) in cardiac tissues from hindlimb unloading (HU) and sham mice. n = 5 mice per group. Data are presented as mean ± SEM. *$p < 0.05$.

Different types of ubiquitin chains determine the functions of ubiquitin chains, much like a precise code governing protein function [34,36]. Previous literatures have reported that changes in the ubiquitination levels of kinases affect the glucose pathway, thereby regulating tumor initiation and inflammatory responses [46–48]. Our analysis revealed that microgravity caused significant changes of non-degradative ubiquitination modifications in kinases of glucose metabolism, such as PGK1, PFKP, and ALDOA. Abnormal ubiquitination always leads to the degradation or accumulation of certain key proteins, resulting in excessive activation of glucose metabolism [49]. Our results suggested an unexpected activation of PGK1, PFKP, and ALDOA by non-degradative ubiquitination under weightless conditions in hearts. Excessive activation of glucose metabolism interferes with normal metabolic signaling pathways, affects cardiomyocytes metabolism, and finally damages heart function [42–44]. And this is a probable explanation for microgravity-induced cardiac dysfunction in our study. At present, some compounds, such as 3-BrPA, significantly inhibit the glycolytic activity, attenuates pulmonary artery remodeling and ventricular hypertrophy, thereby improving heart function [50]. These compounds would be a potential drug for the recovery of microgravity-exposed hearts, worthy of further study and test for cardiac atrophy under spaceflight conditions.

Weightlessness induces functional changes in multiple tissues and organs [51–53]. Based on functional enrichment analysis of differential spectral data from cardiac proteomics and ubiquitination, we found that the upregulation of cardiac proteins due to weightlessness was closely associated with immune response, suggesting the potential regulatory function of immune tissues such as lymph nodes, spleen, bone marrow, and thymus on the heart. Additionally, we found that downregulated proteins were closely related to lipid metabolism, allowing for further detection of the stress response between hearts and tissues such as adipose tissue, liver, intestine, and adrenal cortex to weightlessness. Thus, the analysis of proteomics and ubiquitination data under weightlessness conditions in cardiac tissue will provide further valuable clues for studying the damage caused by weightlessness across the organism.

## Conclusions

In this study, we analyzed the differential proteomic and ubiquitinomic profiles of mouse cardiac tissues exposed to microgravity. In the proteomic profile, upregulated proteins in hearts exposed to microgravity were found to be closely associated with biological functions such as immune response, RNA splicing, and protein folding, while downregulated proteins were linked to cellular scaffolding, lipid metabolism, and energy metabolism, indicating that microgravity affects several crucial biological processes. In contrast, unlike the proteomic data, differences in the ubiquitinomic profile were associated with abnormalities in glucose metabolism pathways. Experimental data revealed significant changes in the ubiquitination of multiple kinases involved in glucose metabolism pathways under microgravity conditions, corroborated by biochemical assays demonstrating abnormal glucose metabolism in the heart. Overall, our findings provide valuable insights into the changes in protein and ubiquitin profiles in cardiac tissue under microgravity conditions. This study may contribute significantly to understanding microgravity-induced cardiac damage and its underlying molecular mechanisms.

## Limits of the study

The limitation of this study is mainly derived from the tail suspension model, which mimics a partial microgravity environment. While the tail suspension model is a valuable tool for studying microgravity's effects in current studies [54–56], it cannot fully replicate the microgravity effects on physiological systems, including muscle atrophy, osteoporosis, decreased immune functions, changes of cardiovascular system, and other physiological response. Therefore, the evidences for our findings should be corroborated by experiments conducted outside Earth orbit in real microgravity conditions.

## Supporting information

**S1 Fig. Ultrasound examination of mouse hearts.** Utilizing echocardiography, measurements of left ventricular dimensions in both systole and diastole were meticulously performed for the mice in the sham group as well as those in the hindlimb unloading (HU) group. Subsequently, the ejection fraction (EF) and fractional shortening (FS) were calculated based on these measurements. n = 5 mice per group. Data were presented as mean ± SEM. ****$p < .0001$.
(TIF)

**S1 Table. Differential proteins identified by proteomic analysis.**
(XLSX)

**S2 Table. Differential ubiquitination sites identified by ubiquitinomic analysis.**
(XLSX)

**S3 Table. Correlation of proteomic and ubiquitinomic data.**
(XLSX)

**S4 Table. Ubi independent mRNA regulated proteins.**
(XLSX)

**S5 Table. Potential mRNA regulated proteins.**
(XLSX)

**S1 Raw image.**
(TIF)

**S1 Raw data.**
(XLSX)

## Author Contributions

**Conceptualization:** Lingqiang Zhang, Yesheng Fu.

**Data curation:** Xin Zhang, Xuemei Zhou, Zhiwei Tu, Lihua Qiang, Zhe Lu, Yesheng Fu.

**Formal analysis:** Xin Zhang, Cui Hua Liu, Lingqiang Zhang, Yesheng Fu.

**Funding acquisition:** Lingqiang Zhang, Yesheng Fu.

**Investigation:** Xin Zhang, Cui Hua Liu, Yesheng Fu.

**Methodology:** Xin Zhang, Xuemei Zhou, Zhiwei Tu, Lihua Qiang, Zhe Lu, Yuping Xie, Yesheng Fu.

**Project administration:** Cui Hua Liu, Lingqiang Zhang, Yesheng Fu.

**Resources:** Xin Zhang, Xuemei Zhou, Lihua Qiang, Yuping Xie, Yesheng Fu.

**Software:** Xin Zhang, Xuemei Zhou, Zhiwei Tu, Yuping Xie, Yesheng Fu.

**Supervision:** Lingqiang Zhang, Yesheng Fu.

**Validation:** Xin Zhang, Yesheng Fu.

**Visualization:** Yesheng Fu.

**Writing – original draft:** Xin Zhang, Yesheng Fu.

**Writing – review & editing:** Lingqiang Zhang, Yesheng Fu.

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
