## [Decision Letter · Decision Letter 0]

16 Aug 2024

PONE-D-24-28980Proteomic and ubiquitinome analysis reveal that microgravity affects cardiac functions by remodeling glucose metabolismPLOS ONE

Dear Dr. Fu,

Thank you for submitting your manuscript to PLOS ONE. After careful consideration, we feel that it has merit but does not fully meet PLOS ONE’s publication criteria as it currently stands. Therefore, we invite you to submit a revised version of the manuscript that addresses the points raised during the review process.

The manuscript provides a detailed analysis of the proteomic and ubiquitinomic alterations in mouse cardiac tissues exposed to microgravity, attempting to elucidate molecular mechanisms underlying cardiac dysfunction. However, there are several critical areas that require significant revision or clarification before the study can be considered for publication.

**Methodological Limitations:** While the study uses proteomic and ubiquitinomic analyses, it fails to account for potential confounding variables in the experimental setup. For instance, the tail suspension model used to simulate microgravity is well known but has limitations in accurately mimicking the conditions experienced in space. This raises concerns about the validity and translatability of the findings to actual spaceflight conditions. The authors should address how they intend to mitigate these limitations or at least provide a more robust discussion of the model’s limitations.**Data Presentation and Clarity:** The figures and tables provided, while comprehensive, are often confusing. For example, Figure 1 displays proteomic analysis but lacks sufficient annotations for non-expert readers to fully understand the significance of the data. The legends accompanying the figures are vague and do not provide enough information regarding statistical analyses or controls used. This weakens the overall impact of the presented data and calls into question the validity of the results.**Lack of Contextual Integration:** The authors discuss cardiac dysfunction in a microgravity environment but fail to integrate similar relevant studies effectively into the introduction. For example, recent findings such as those by Al-Awaida et al. (2020, 2023) on the effects of simulated microgravity on the biological properties of wheatgrass and its therapeutic effects on diabetes and breast cancer under microgravity conditions could provide valuable context. These studies have demonstrated that microgravity can significantly modulate metabolic and physiological responses in mammalian models. Including such literature would greatly enhance the introduction by providing a broader perspective on how microgravity affects not only cardiac function but also other biological systems.

To strengthen the introduction, it is strongly recommended that the following articles be cited:

Al-Awaida, Wajdy J., et al. "Effect of simulated microgravity on the antidiabetic properties of wheatgrass (Triticum aestivum) in streptozotocin-induced diabetic rats." npj Microgravity 6.1 (2020): 6.Al-Awaida, Wajdy, et al. "Modulation of wheatgrass (Triticum aestivum Linn) toxicity against breast cancer cell lines by simulated microgravity." Current Research in Toxicology 5 (2023): 100127.**Inadequate Interpretation of Ubiquitinome Results:** The interpretation of the ubiquitinome data lacks depth. While the authors claim that non-degradative ubiquitination plays a role in metabolic dysfunction, they do not provide sufficient mechanistic insights or experimental validation to support this claim. For instance, the observed changes in kinase activity related to glucose metabolism are not fully explained in terms of their physiological relevance or implications for cardiac dysfunction. Furthermore, there is little discussion on how these findings could inform therapeutic interventions or countermeasures for cardiac atrophy under spaceflight conditions.**Conclusions Exceed the Evidence:** The conclusions drawn in the paper are overly broad and do not sufficiently align with the data presented. Specifically, the claim that the study provides "valuable insights for long-term human space exploration" is premature, given the limitations in experimental design and lack of direct spaceflight data. The authors need to scale back their conclusions and provide a more conservative interpretation of their findings until they are validated by further research, possibly in actual spaceflight conditions.

In summary, this manuscript has the potential to contribute valuable information to the field of space medicine. However, significant revisions are required in methodology, data presentation, and interpretation. The integration of relevant literature and a clearer discussion of the limitations and broader implications are essential to elevate the study to the level of publication.

We look forward to receiving your revised manuscript.

Kind regards,

Wajdy Jum’ah Al-Awaida, Ph.D

Academic Editor

PLOS ONE

3. PLOS requires an ORCID iD for the corresponding author in Editorial Manager on papers submitted after December 6th, 2016. Please ensure that you have an ORCID iD and that it is validated in Editorial Manager. To do this, go to ‘Update my Information’ (in the upper left-hand corner of the main menu), and click on the Fetch/Validate link next to the ORCID field. This will take you to the ORCID site and allow you to create a new iD or authenticate a pre-existing iD in Editorial Manager. Please see the following video for instructions on linking an ORCID iD to your Editorial Manager account: https://www.youtube.com/watch?v=_xcclfuvtxQ".

4. To comply with PLOS ONE submissions requirements, in your Methods section, please provide additional information regarding the experiments involving animals and ensure you have included details on (1) methods of sacrifice, (2) efforts to alleviate suffering.

5. We suggest you thoroughly copyedit your manuscript for language usage, spelling, and grammar. If you do not know anyone who can help you do this, you may wish to consider employing a professional scientific editing service. 

A clean copy of the edited manuscript (uploaded as the new *manuscript* file)”.

6. We note that the grant information you provided in the ‘Funding Information’ and ‘Financial Disclosure’ sections do not match. 

7. Thank you for stating the following financial disclosure: 

 [This work was supported by National Natural Science Foundation of China (82192881, 32201023), National Key Research and Development Project of China (2021YFA1300200), Young Elite Scientists Sponsorship Program by CAST (YESS20220049) ].  

8. Thank you for stating the following in the Acknowledgments Section of your manuscript: 

[This work was supported by National Natural Science Foundation of China (82192881, 32201023), National Key Research and Development Project of China (2021YFA1300200), Young Elite Scientists Sponsorship Program by CAST (YESS20220049) ]

 [This work was supported by National Natural Science Foundation of China (82192881, 32201023), National Key Research and Development Project of China (2021YFA1300200), Young Elite Scientists Sponsorship Program by CAST (YESS20220049) ]

9. We notice that your supplementary figures are uploaded with the file type 'Figure'. Please amend the file type to 'Supporting Information'. Please ensure that each Supporting Information file has a legend listed in the manuscript after the references list.

Reviewers' comments:

Reviewer's Responses to Questions

**Comments to the Author**

1. Is the manuscript technically sound, and do the data support the conclusions?

Reviewer #1: Yes

Reviewer #2: Yes

2. Has the statistical analysis been performed appropriately and rigorously? 

Reviewer #1: Yes

Reviewer #2: Yes

3. Have the authors made all data underlying the findings in their manuscript fully available?

Reviewer #1: Yes

Reviewer #2: Yes

4. Is the manuscript presented in an intelligible fashion and written in standard English?

Reviewer #1: Yes

Reviewer #2: Yes

5. Review Comments to the Author

Reviewer #1: A good study which tries to combine biochemical hypothesis and clinical parameters on echo. M mode is not only way to calculate EF. secondly its operator dependent. this low EF by M mode may be due to procedure related. but overall good study

Reviewer #2: Hello,

Thanks for submitting manuscript titled "Proteomic and ubiquitinome analysis reveal that microgravity affects cardiac functions by remodeling glucose metabolism"to this journal.

1. It is an interesting article trying to assess the same in MICE and not humans. No where in either the title or abstract this is clarified as to what subgroup of population was evaluated. Though the proteins changes can be extrapolated across the species but the proof of the same lies on the investigator and hence it would be important to mention Mice derived proteins rather than a blanket statement.

2. The methods and ethical considerations needs to be elaborated further in the mansucript.

3. there are minor grammatical and typographical errors which need to be rectified by a native English speaking subject expert.

Otherwise it is a well written draft and can be accepted after the revision.

Thanks

6. PLOS authors have the option to publish the peer review history of their article (what does this mean?). If published, this will include your full peer review and any attached files.

Reviewer #1: **Yes: **Syed Fayaz Mujtaba

Reviewer #2: **Yes: **Dr Kamal H sharma

---

## [Author Response · Author response to Decision Letter 0]

25 Oct 2024

Response to Reviewers' Comments：

We sincerely thank the two reviewers for their evaluation of this paper and their recognition of it. Below are our responses to the reviewers' questions.

Reviewer #2: Hello,

Thanks for submitting manuscript titled "Proteomic and ubiquitinome analysis reveal that microgravity affects cardiac functions by remodeling glucose metabolism" to this journal.

1. It is an interesting article trying to assess the same in MICE and not humans. No where in either the title or abstract this is clarified as to what subgroup of population was evaluated. Though the proteins changes can be extrapolated across the species but the proof of the same lies on the investigator and hence it would be important to mention Mice derived proteins rather than a blanket statement.

2. The methods and ethical considerations needs to be elaborated further in the manuscript.

3. there are minor grammatical and typographical errors which need to be rectified by a native English speaking subject expert.

Otherwise, it is a well written draft and can be accepted after the revision.

Thanks

Response: We are very grateful for the valuable suggestions put forward by the reviewers. We have modified the title, supplemented and improved the methods and ethics in the manuscript, and also re-proofread the manuscript. We sincerely appreciate your recognition of this research.

---

## [Editor Report · Decision Letter 1]

28 Oct 2024

Proteomic and ubiquitinome analysis reveal that microgravity affects glucose metabolism of mouse hearts by remodeling non-degradative ubiquitination

PONE-D-24-28980R1

Dear Dr. Fu,

We’re pleased to inform you that your manuscript has been judged scientifically suitable for publication and will be formally accepted for publication once it meets all outstanding technical requirements.

Kind regards,

Wajdy Jum’ah Al-Awaida, Ph.D

Academic Editor

PLOS ONE

Additional Editor Comments (optional):

The revised manuscript, "Proteomic and ubiquitinome analysis reveal that microgravity affects glucose metabolism of mouse hearts by remodeling non-degradative ubiquitination," has undergone significant improvements in response to reviewers' insightful feedback. The title was refined to specify the study’s focus on glucose metabolism in mouse hearts, enhancing clarity. Methodological details, particularly ethical considerations, were expanded to provide greater transparency. Background information on microgravity’s physiological impacts was integrated into the introduction to strengthen context. The discussion now includes additional literature, deepening the analysis of proteomic and ubiquitination changes. A new "Limitations of the Study" section was added in the conclusions, addressing the limitations of the simulated microgravity model and aligning the manuscript with reviewers' suggestions for greater scientific depth and precision
---

## [Editor Report · Acceptance letter]

5 Nov 2024

PONE-D-24-28980R1 

PLOS ONE

Dear Dr. Fu, 

I'm pleased to inform you that your manuscript has been deemed suitable for publication in PLOS ONE. Congratulations! Your manuscript is now being handed over to our production team.

Kind regards, 

on behalf of

Prof. Wajdy Jum’ah Al-Awaida 

Academic Editor

PLOS ONE